# Multivariate and Multiscale Complexity of Long-Range Correlated Cardiovascular and Respiratory Variability Series

**DOI:** 10.3390/e22030315

**Published:** 2020-03-11

**Authors:** Aurora Martins, Riccardo Pernice, Celestino Amado, Ana Paula Rocha, Maria Eduarda Silva, Michal Javorka, Luca Faes

**Affiliations:** 1Faculdade de Ciências, Universidade do Porto, Rua Campo Alegre, 4169-007 Porto, Portugal; aurora.ramalho@hotmail.com (A.M.); celestino.amado@gmail.com (C.A.); aprocha@fc.up.pt (A.P.R.); 2Centro de Matemática da Universidade do Porto (CMUP), 4169-007 Porto, Portugal; 3Department of Engineering, University of Palermo, Viale delle Scienze, Bldg. 9, 90128 Palermo, Italy; luca.faes@unipa.it; 4Faculdade de Economia, Universidade do Porto, Rua Dr. Roberto Frias, 4169-007 Porto, Portugal; mesilva@fep.up.pt; 5Centro de Investigação e Desenvolvimento em Matemática e Aplicações (CIDMA); 6Department of Physiology, Comenius University in Bratislava, Jessenius Faculty of Medicine, Mala Hora 4C, 03601 Martin, Slovakia; michal.javorka@uniba.sk; 7Biomedical Center Martin, Comenius University in Bratislava, Jessenius Faculty of Medicine, Mala Hora 4C, 03601 Martin, Slovakia

**Keywords:** multi-scale entropy (MSE), vector autoregressive fractionally integrated (VARFI) models, heart rate variability (HRV), systolic arterial pressure (SAP)

## Abstract

Assessing the dynamical complexity of biological time series represents an important topic with potential applications ranging from the characterization of physiological states and pathological conditions to the calculation of diagnostic parameters. In particular, cardiovascular time series exhibit a variability produced by different physiological control mechanisms coupled with each other, which take into account several variables and operate across multiple time scales that result in the coexistence of short term dynamics and long-range correlations. The most widely employed technique to evaluate the dynamical complexity of a time series at different time scales, the so-called multiscale entropy (MSE), has been proven to be unsuitable in the presence of short multivariate time series to be analyzed at long time scales. This work aims at overcoming these issues via the introduction of a new method for the assessment of the multiscale complexity of multivariate time series. The method first exploits vector autoregressive fractionally integrated (VARFI) models to yield a linear parametric representation of vector stochastic processes characterized by short- and long-range correlations. Then, it provides an analytical formulation, within the theory of state-space models, of how the VARFI parameters change when the processes are observed across multiple time scales, which is finally exploited to derive MSE measures relevant to the overall multivariate process or to one constituent scalar process. The proposed approach is applied on cardiovascular and respiratory time series to assess the complexity of the heart period, systolic arterial pressure and respiration variability measured in a group of healthy subjects during conditions of postural and mental stress. Our results document that the proposed methodology can detect physiologically meaningful multiscale patterns of complexity documented previously, but can also capture significant variations in complexity which cannot be observed using standard methods that do not take into account long-range correlations.

## 1. Introduction

Cardiovascular oscillations are influenced by the combined activity of different physiological regulation processes and, as a consequence, exhibit a rich dynamical structure [1]. Such different processes do not usually work at a single time-scale, but instead operate across different temporal scales, that for example reflect thermoregulatory or neural parasympathetic and sympathetic control. For this reason, different methods have been recently developed to evaluate the ’multiscale complexity’ of cardiovascular oscillations. These methods allow both to characterize the physiological regulatory systems and to extract diagnostic parameters, and thus can have noteworthy clinical implications: in fact, a decrease of dynamical complexity can be related to an impaired capability of the subsystems composing the organism to interact among each other and it has been already proposed as a marker of pathological conditions [2,3]. Among the proposed methods, the one which is likely most popular is the so-called multiscale entropy (MSE), developed by Costa et al. [4]. This method calculates the conditional entropy (CE) of a single time series (usually the heart period, HP) as a function of the time scale of observation; the change of time scale is achieved through averaging consecutive segments of the time series via a procedure that has been lately recognized to correspond to a two-step process consisting of a low-pass filter followed by downsampling [5]. It is worth noting, however, that the initial formulation of multiscale entropy suffered from drawbacks related both to issues due to the filtering and downsampling steps, and to the unsuitability of CE analysis in conditions of data paucity caused by the availability of short time series and by the needs to explore multivariate time series at coarse time scales. Therefore, in the last years, the definition of MSE has been refined to take into account typical requirements of cardiovascular and cardiorespiratory signal analysis, and specifically: (i) to allow the joint calculation of complexity of multiple variables besides HP, for example systolic arterial pressure (SAP) and respiration (RESP) [6]; (ii) to allow the assessment of the complexity of shorter time series, usually few hundred beats long [7,8]. For short-term physiological time series, complexity has been related to the regularity of the temporal patterns observed in the signals, and thus is usually a measure of the unpredictability of the present sample given a limited number of past samples [9]. However, recent studies have recognized the importance of long-range correlations resulting in slowly varying dynamics also for the analysis of short-term complexity [10], and have started to account for these correlations in multiscale entopy-based analysis [8].

In this work, we introduce a novel method to compute multivariate and multiscale complexity of cardiovascular oscillations. This method fits a multivariate time series using a vector autoregressive fractionally integrated (VARFI) model and then provides the multiscale representation of the VARFI parameters using the theory of state space models, allowing to extract from such parameters multiscale and multivariate measures of complexity. This approach presents several advantages if compared to other previous works in the same field [4,5,6,7,8,10], and in particular: (i) the parametric formulation employed permits to work reliably on short time series; (ii) fractional integration allows to take into account not only short-term dynamics, but also the long-range correlations; (iii) the vector formulation permits to compute the overall complexity of multivariate time series, or the complexity of an individual time series when one or more other series are considered. In this work, such approach is applied to HP, SAP and RESP time series measured in healthy subjects monitored in a resting condition and during two types of physiological stress: postural stress provoked by head-up tilt and mental stress induced by mental arithmetics.

The algorithms for multivariate multiscale analysis of physiological time series presented in this work are collected in the MSE-VARFI MATLAB toolbox, which can be freely downloaded from http://www.lucafaes.net/LMSE-MSE_VARFI.html.

## 2. Methods

### 2.1. Measures of Complexity in Linear Multivariate Stochastic processes

Considering a dynamical system X whose activity is defined by the zero-mean stationary multivariate stochastic process $X=[X_{1}\cdots X_{M}]$, where each $X_{j}$, j=1,⋯,M, denotes a scalar stochastic process $X_{j}=\left[\cdots X_{j,1}\cdots X_{j,n}\cdots \right]$, let us define as $X_{n}=\left[X_{1,n}\cdots X_{M,n}\right]$ the M-dimensional random variable describing the present state of the system, and as $X_{n}^{-}=\left[X_{n-1}X_{n-2}\ldots \right]$ the infinite-dimensional vector variable describing its past states. Then, a measure of complexity of the system, typically defined for univariate systems [10] and here extended to the multivariate case, is the entropy rate defined as
(1)$$C_{X}=H\left(X_{n}|X_{n}^{-}\right)=H\left(X_{n+1}^{-}\right)-H\left(X_{n}^{-}\right),$$
where $H(X_{n}|X_{n}^{-})$ is the conditional entropy of the present given the past, with H(·) denoting the Shannon Entropy. If the observed process X is a jointly distributed Gaussian process, it can be described without loss of generality through a vector linear regression model fed by white and uncorrelated innovations $E_{n}=\left[E_{1,n}\cdots E_{M,n}\right]$ such that, for each j∈1,…,M, $E_{j,n}=X_{j,n}-E\left[X_{j,n}|X_{n}^{-}\right]$ [11]. In such a case, the entropy of the present state of the process and the conditional entropy of the present given the past can be expressed analytically in terms of the covariance of the process, $\Sigma_{X}=E\left[X_{n}^{T}X_{n}\right]$, and the covariance of the innovations, $\Sigma_{E}=E\left[E_{n}^{T}E_{n}\right]$, as [11,12,13]:
(2a)$$\begin{array}{c} H\left(X_{n}\right)=\frac{1}{2}\ln({\left(2\pi e\right)}^{M}|\Sigma {}_{X}\left|\right), \end{array}$$
(2b)$$\begin{array}{c} H\left(X_{n}|X_{n}^{-}\right)=\frac{1}{2}\ln({\left(2\pi e\right)}^{M}|\Sigma {}_{E}\left|\right), \end{array}$$
where |·| stands for matrix determinant. Then, it follows immediately that the complexity of a multivariate linear process with joint Gaussian distribution is given by Equation (2b). In this work we provide an alternative definition of complexity, which includes a normalization of the innovation covariance to the process covariance:(3)$${\bar{C}}_{X}=\frac{1}{2}\ln\left({\left(2\pi e\right)}^{M}\frac{|\Sigma_{E}|}{|\Sigma_{X}|}\right).$$Such normalization, which is implicitly implemented in studies assessing the complexity of univariate time series where the series is normalized to unit variance before computing the complexity measure, is formalized here in Equation (Equation 3) for multivariate series, and will be fundamental in the definition of multiscale complexity where the process covariance intrinsically changes with the time scale.

The measure of global complexity defined above for a multivariate process can be particularized to the characterization of one of its constituent processes. Specifically, the complexity of a scalar process $X_{j}\in X$, j∈1,…M, can be defined in an univariate context with respect to its own dynamics only, in a bivariate context with respect to its dynamics and to the dynamics of another scalar process $X_{i}\in X,i\neq j$, or in a fully multivariate context with respect to the dynamics of the whole observed vector process X [14]. The derivations are based on the knowledge that, in Gaussian processes, a linear parametric representation captures all of the entropy differences that define the conditional entropies [11] and that these entropy differences are related to the partial variances of the present of the target given its past and the past of one or more other processes, intended as variances of the prediction errors resulting from linear regression [13,15]. Specifically, let us denote as $E_{j|j,n}=X_{j,n}-E\left[X_{j,n}|X_{j,n}^{-}\right]$ and $E_{j|ij,n}=X_{j,n}-E\left[X_{j,n}|X_{i,n}^{-},X_{j,n}^{-}\right]$ the prediction error of a linear regression of $X_{j,n}$ performed respectively on $X_{j,n}^{-}$ and $\left(X_{j,n}^{-},X_{i,n}^{-}\right)$, and consider that the prediction error of a linear regression of $X_{j,n}$ on $X_{n}^{-}$ is the $j^{th}$ innovation process $E_{j,n}$ defined above. Then, denoting the variances of the three prediction errors as $\Sigma_{E_{j|j}}=E\left[E_{j|j,n}^{2}\right]$, $\Sigma_{E_{j|ij}}=E\left[E_{j|ij,n}^{2}\right]$ and $\Sigma_{E_{j}}=E\left[E_{j,n}^{2}\right]=\Sigma_{E}\left(j,j\right)$, the univariate, bivariate and multivariate normalized complexity measures relevant to the process $X_{j}$ are defined as:
(4a)$$\begin{array}{c} {\bar{C}}_{X_{j}|X_{j}}=\frac{1}{2}\ln2\pi e\frac{\Sigma_{E_{j|j}}}{\Sigma_{X_{j}}}, \end{array}$$
(4b)$$\begin{array}{c} {\bar{C}}_{X_{j}|X_{i},X_{j}}=\frac{1}{2}\ln2\pi e\frac{\Sigma_{E_{j|ij}}}{\Sigma_{X_{j}}}, \end{array}$$
(4c)$$\begin{array}{c} {\bar{C}}_{X_{j}|X}=\frac{1}{2}\ln2\pi e\frac{\Sigma_{E_{j}}}{\Sigma_{X_{j}}}, \end{array}$$
where $\Sigma_{X_{j}}=\Sigma_{X}\left(j,j\right)$ is the $j^{th}$ diagonal element of $\Sigma_{X}$.

### 2.2. Linear Multivariate Stochastic Processes with Long Range Correlations

In this section we present the parametric approach to the description of linear multivariate Gaussian stochastic processes exhibiting both short-term dynamics and long-range correlations. The approach is based on representing the *M*-dimensional discrete-time, zero-mean and unit variance stochastic process $X_{n}$ defined in Section 2.1 as a vector autoregressive fractionally integrated (VARFI) process fed by the uncorrelated Gaussian innovations $E_{n}$. The VARFI process takes the form [16]:(5)$$A\left(L\right)\mathrm{diag}\left(\nabla^{d}\right)X_{n}=E_{n},$$
where *L* is the back-shift operator $\left(L^{i}X_{n}=X_{n-i}\right)$, $A\left(L\right)=I_{M}-\sum_{i=1}^{p}A_{i}L^{i}$ ($I_{M}$ is the identity matrix of size *M*) is a vector autoregressive (VAR) polynomial of order *p* defined by the M×M coefficient matrices $A_{1},\ldots ,A_{p}$, and
$$\mathrm{diag}\left(\nabla^{d}\right)=\left[\begin{array}{cccc} {\left(1-L\right)}^{d_{1}} & 0 & \cdots & 0\\ 0 & {\left(1-L\right)}^{d_{2}} & \cdots & 0\\ \vdots & \vdots & \; & \vdots \\ 0 & 0 & \cdots & {\left(1-L\right)}^{d_{M}} \end{array}\right],$$
where ${\left(1-L\right)}^{d_{i}},i=1,\cdots ,M$, is the fractional differencing operator defined by [17]:(6)$${\left(1-L\right)}^{d_{i}}=\sum_{k=0}^{\infty }G_{k}^{\left(i\right)}L^{k}\;,\;G_{k}^{\left(i\right)}=\frac{\Gamma (k-d_{i})}{\Gamma \left(-d_{i}\right)\Gamma \left(k+1\right)},$$
with Γ(·) denoting the gamma (generalized factorial) function. The parameter $d=(d_{1},\ldots ,d_{M})$ in Equation (Equation 5) determines the long-term behavior of each individual process, while the coefficients of the polynomial A(L) allow description of the short-term dynamics. Note that the process defined in Equation (Equation 5) is a particular case of the broader class of VARFIMA(p,d,l) processes [16], which also contains the class of autoregressive processes VAR(p); here we restrict our analysis to the description of the VARFIMA(p,d,0) process, which we denote as a VARFI(p,d) process.

The parameters of the VARFI model (Equation 5), namely the coefficients of A(L) and the variance of the innovations $\Sigma_{E}$, are typically obtained from process realizations of finite length first estimating the differencing parameters $d_{i}$ by means of the Whittle semi-parametric local estimator [17] separately for each individual process $X_{i}$, then defining the filtered data $X_{i,n}^{\left(f\right)}={\left(1-L\right)}^{d_{i}}X_{i,n}$, and finally estimating the VAR parameters from the filtered data $X_{n}^{\left(f\right)}$ using the ordinary least squares method to solve the VAR model $A\left(L\right)X_{n}^{\left(f\right)}=E_{n}$, with model order *p* assessed through the Bayesian information criterion [18].

### 2.3. Multiscale Complexity of VARFI Processes

In this section we report how to compute across multiple temporal scales the complexity measures defined in Section 2.1, under the hypothesis that the analyzed multivariate process is properly described by the VARFI representation provided in Section 2.2. The procedure for multiscale complexity analysis, which extends the approach proposed in Reference [19] to multivariate processes incorporating long-range correlations, is presented here reporting the essential steps and described with more mathematical details in the Appendices. There are three appendices to the main text. Appendix A reports the procedure for computing the coefficients of a finite-order VAR process that approximates an assigned VARFI process. Appendix B recalls the derivations relevant to the identification of multiscale state space (SS) processes defined as filtered and downsampled versions of an assigned VAR process; the parameters of the SS process defined at an assigned time scale are exploited to compute the multivariate complexity measure at that time scale. Appendix C describes how to define restricted SS processes and to rearrange them in order to extract the partial variances needed for the computation of the univariate and bivariate multiscale complexity measures. The derivations described in Appendix B and Appendix C are taken from Refs. [7,8,19,20].

Before implementing the rescaling procedure, we approximate the VARFI process (Equation 5) with a finite order VAR process by truncating the fractional integration part at a finite lag *q*, that is, we perform the following approximation:
(7a)$$\begin{array}{c} \mathrm{diag}\left(\nabla^{d}\right)\approx G\left(L\right)=\sum_{k=0}^{q}G_{k}L^{k}, \end{array}$$
(7b)$$\begin{array}{c} G_{k}=\mathrm{diag}\left[G_{k}^{\left(1\right)},\cdots ,G_{k}^{\left(M\right)}\right]. \end{array}$$This allows us to rewrite the VARFI(p,d) process as a VAR(*m*) process, where m=p+q:(8)$$B\left(L\right)X_{n}=E_{n},$$
where the new coefficients are B(L)=A(L)G(L), with A(L) as in Equation (Equation 5) and G(L) as in Equation ([Disp-formula FD7a-entropy-22-00315]). The exact procedure to derive the coefficients of the VAR(*m*) process is explained in Appendix A.

In order to represent a scalar stochastic process at the temporal scale defined by the scale factor τ, a two step procedure is typically employed which consists first in filtering the process with a lowpass filter with cutoff frequency $f_{\tau }=1/\left(2\tau \right)$, and then downsampling the filtered process using a decimation factor τ [5,19]. Extending this approach to the multivariate process X, we first implement the following linear finite impulse response (FIR):(9)$$\begin{array}{c} X_{n}^{\left(r\right)}=D\left(L\right)X_{n}\;, \end{array}$$
where *r* denotes the filter order and $D\left(L\right)=\sum_{k=0}^{r}I_{M}D_{k}L^{k}$, where the coefficients of the polynomial $D_{k}$, k=1,…,r, are the same for all scalar processes $X_{j}\in X$ and are chosen to set up a lowpass FIR configuration with cutoff frequency 1/(2τ). The filtering step transforms the VAR(*p+q*) process (Equation 8) into a VARMA(*p+q,r*) process with moving average (MA) part determined by the filter coefficients:(10)$$\begin{array}{c} B\left(L\right)X_{n}^{\left(r\right)}=D\left(L\right)B\left(L\right)X_{n}=D\left(L\right)E_{n}\;. \end{array}$$Then, we exploit the connection between VARMA processes and state space (SS) processes [21] to evidence that the VARMA process (Equation 10) can be expressed in SS form as:
(11a)$$\begin{array}{cc} Z_{n+1}^{\left(r\right)} & =B^{\left(r\right)}Z_{n}^{\left(r\right)}+K^{\left(r\right)}E_{n}^{\left(r\right)} \end{array}$$
(11b)$$\begin{array}{cc} X_{n}^{\left(r\right)} & =C^{\left(r\right)}Z_{n}^{\left(r\right)}+E_{n}^{\left(r\right)}, \end{array}$$
where $Z_{n}^{\left(r\right)}={\left[X_{n-1}^{\left(r\right)}\cdots X_{n-m}^{\left(r\right)}E_{n-1}\cdots E_{n-r}\right]}^{T}$ is a (m+r)-dimensional state process, $E_{n}^{\left(r\right)}=D_{0}E_{n}$ is the SS innovation process, and the vectors $K^{\left(r\right)}$ and $C^{\left(r\right)}$ and the matrix $B^{\left(r\right)}$ can be obtained from B(L) and D(L) (see Appendix B).

The second step of the rescaling procedure is to downsample the filtered process in order to complete the multiscale representation. This is done exploiting theoretical findings [7,22,23] which allow to describe the filtered SS process after downsampling in the form:
(12a)$$\begin{array}{cc} Z_{n+1}^{\left(\tau \right)} & =B^{\left(\tau \right)}Z_{n}^{\left(\tau \right)}+K^{\left(\tau \right)}E_{n}^{\left(\tau \right)} \end{array}$$
(12b)$$\begin{array}{cc} X_{n}^{\left(\tau \right)} & =C^{\left(\tau \right)}Z_{n}^{\left(\tau \right)}+E_{n}^{\left(\tau \right)}. \end{array}$$Equation (12) provides the SS form of the filtered and downsampled version of the original VARFI(p,d) process, and its parameters ($B^{\left(\tau \right)},C^{\left(\tau \right)},K^{\left(\tau \right)},\Sigma_{E^{\left(\tau \right)}}$) can be obtained from the SS parameters before downsampling and from the downsampling factor τ; moreover, the variance of the downsampled process, $\Sigma_{X^{\left(\tau \right)}}$, can be computed analytically from the parameters of the SS model (12) by solving a discrete-time Lyapunov equation (these steps are shown in the Appendix B). The parameters relevant to the computation of complexity at scale τ are the variance of the downsampled process, $\Sigma_{X^{\left(\tau \right)}}$, and the variance of the corresponding innovations, $\Sigma_{E^{\left(\tau \right)}}$. These variances can indeed be combined in a similar way to that of Equation (Equation 3) to yield the expression of the complexity of the original process $X_{n}$ when it is observed at scale τ:(13)$${\bar{C}}_{X}=\frac{1}{2}\ln\left({\left(2\pi e\right)}^{M}\frac{|\Sigma_{E^{\left(\tau \right)}}|}{|\Sigma_{X^{\left(\tau \right)}}|}\right).$$Note that this measure tends to the theoretical value $\frac{1}{2}\ln{\left(2\pi e\right)}^{M}$ when τ→∞.

Finally, we show how to compute any partial variance appearing in Equation (4) from the parameters of an SS model in the form of (12), so that the three complexity measures relevant to the scalar process $X_{j}$ can be computed at any assigned time scale τ. To do this, we exploit the formulations reported in Refs. [22,23], showing that the partial variance $\Sigma_{E_{j|a}^{\left(\tau \right)}}$, where the subscript *a* denotes any combination of indexes ∈1,⋯,M, can be derived from the SS representation of the innovations of a submodel obtained removing the variables not indexed by *a* from the observation equation. Specifically, we need to consider the submodel with state Equation ([Disp-formula FD12a-entropy-22-00315]) and observation equation:(14)$$X_{a,n}^{\left(\tau \right)}=C_{a}^{\left(\tau \right)}Z_{n}^{\left(\tau \right)}+E_{a,n}^{\left(\tau \right)},$$
where the additional subscript ${}_{a}$ denotes the selection of the rows with indices *a* in a vector or a matrix. This restricted SS model can be rearranged to extract the partial variance $\Sigma_{E_{j|a}^{\left(\tau \right)}}$ from the covariance matrix of its innovations (see Appendix C), so that with this procedure the univariate, bivariate and multivariate normalized complexity measures can be computed inserting the partial variances $\Sigma_{E_{j|j}^{\left(\tau \right)}}$, $\Sigma_{E_{j|ij}^{\left(\tau \right)}}$ in Equation ([Disp-formula FD4a-entropy-22-00315]) and Equation (4b), and using $\Sigma_{E_{j}}^{\left(\tau \right)}=\Sigma_{E^{\left(\tau \right)}}\left(j,j\right)$ in Equation (4c), together with $\Sigma_{X_{j}}^{\left(\tau \right)}=\Sigma_{X^{\left(\tau \right)}}\left(j,j\right)$ from Equation (A9b). These individual measures tend to the theoretical value $\frac{1}{2}\ln\left(2\pi e\right)$ when τ→∞.

## 3. Application to Cardiovascular Variability Processes

In this section the proposed method is illustrated using cardiovascular and respiratory series: the heart period (HP), systolic arterial pressure (SAP), and respiration (RESP). Several studies report the interaction between the dynamics of these three time series [24,25,26,27,28], which motivates their use in a multivariate context. The variation of heart period, usually referred as heart rate variability (HRV), reflecting cardiovascular complexity and representing the capability of the organism to react to environmental and psychological stimuli, is the most studied variable and main target variable in cardiovascular and cardiorespiratory spontaneous variability [29,30,31]. For this reason, the conditional measures of a single scalar process will focus on the HP time series, as it has been shown that SAP and RESP have an effect (direct or indirectly) on this process [24,25,26,27,28].

### 3.1. Experimental Protocol

The HP, SAP and RESP time series were measured in a group of 62 healthy subjects (19.5±3.3 years old, 37 females) monitored in the resting supine position (SU${}_{1}$), in the upright position (UP) reached through passive head-up tilt, in the recovery supine position (SU${}_{2}$) and during mental stress induced by mental arithmetics (MA) [32]. During the measurements, the subjects were free-breathing. For each subject and condition, the multivariate process X is defined as $X=[X_{HP},X_{SAP},X_{RESP}]$. The acquired signals were the surface electrocardiogram (ECG), the finger arterial blood pressure recorded noninvasively by the photoplethysmographic method, and the respiration signal recorded through respiratory inductive plethysmography. For each subject and experimental condition, the values of HP, SAP and RESP were measured on a beat-to-beat basis respectively as the sequences of the temporal distances between consecutive R peaks of the ECG, the maximum values of the arterial pressure waveform taken within the consecutively detected heart periods, and the values of the respiratory signal sampled at the onset of the consecutively detected heart periods. The study was approved by Ethical Committee of the Jessenius Faculty of Medicine, Comenius University and all participants signed a written informed consent. A detailed description of the experimental protocol and signal measurement is reported in Ref. [32].

The analysis was performed on segments of at least 400 consecutive points, free of artifacts and deemed as weak-sense stationary through visual inspection, extracted from the time series for each subject and condition. Missing values and outliers were corrected through linear interpolation and, for HP and when possible, erroneous/missing intervals were substituted by pulse intervals measured as the difference in time between two consecutive SAP measurements ($\Delta_{t_{SAP}}\left(n\right)=t_{SAP}\left(n+1\right)-t_{SAP}\left(n\right)$). The three time series were normalized to zero mean and unit variance before multiscale analysis.

### 3.2. Data Analysis

Two different approaches were followed to compute multiscale complexity: (i) the “eVAR” approach, based on pure VAR model identification, that is, performing the whole procedure described in Section 2 after forcing d=[0,0,0] in Equation (Equation 5); (ii) the “eVARFI” approach, based on complete VARFI model identification, that is, following the whole procedure described in Section 2 with $d_{i}$, i=1,⋯,3, estimated individually from the original time series and considered in the computations. Pursuing these approaches we compare, respectively, (i) the traditional complexity analysis where long-range correlations are neither removed nor modeled, and (ii) the complexity analysis performed by modeling the long-range correlations and considering them together with the short-term dynamics. Such a comparison is meant to infer the role of long-range correlations in contributing to the multiscale complexity and to its variation between conditions. The VARFI model fitting the time series X was identified first estimating the fractional differencing parameter $d_{i}$, i=1,⋯,3, individually for each time series using the Whittle estimator, then filtering the time series with the fractional integration polynomial truncated at a lag q=50, and finally estimating the parameters of the polynomial relevant to the short-term dynamics through least squares VAR identification. The value of *q* has to be selected to approximate the VARFI process, which is theoretically of infinite order, with a finite order VAR process. According to previous studies [8,33] *q* = 50 is an appropriate value for truncating the VARFI process. By increasing *q*, we can obtain a more precise approximation of the fractional integration part but with a higher computational cost, while a reduced value (and thus an excessive truncation) causes an underestimation of the complexity and the smoothing of the nonmonotonic trends with the timescale [8]. The VAR model order *p* was selected as the minimum of the Bayesian information criterion (BIC) figure of merit [34]. Then, multiscale complexity measures were computed implementing a FIR lowpass filter of order r=48, for time scales τ in the range (1,...,30), which corresponds to lowpass cutoff normalized frequencies $f_{\tau }=\left(0.5,\cdots ,0.01667\right)$. The order of the FIR filter determines its selectivity around the cutoff frequency. In this study, an order *r* = 48 was selected according to previous settings [8]. The changes in complexity related to the oscillatory rhythms are typically evaluated in cardiovascular variability in low-frequency (LF, from 0.04 Hz to 0.15 Hz) and high-frequency (HF, from 0.15 Hz to 0.4 Hz) spectral bands [29]. To account for the different mean heart rate for each subject and condition, multiscale analysis was performed determining the cutoff frequency of the rescaling filter based on the Nyquist frequency in Hz of the HP time series, which for the time scale τ is given by $f_{\tau_{Hz}}=1/\left(2\tau \bar{HP}\right)$, where $\bar{HP}$ stands for the mean of the HP process. In this work, considering the physiological LF and HF frequency ranges, four cutoff frequencies $f_{Hz}$ were chosen to perform the analysis: 0.4 Hz, 0.15 Hz, 0.1 Hz and 0.04 Hz. To obtain the complexity values at such frequencies, linear interpolation was performed on the profiles $f_{\tau_{Hz}}$.

The differencing parameters $d_{i}$ were estimated individually for each time series in the interval [−0.5,1] since the VARFI model is stationary for $-0.5<d_{i}<0.5$, while it is nonstationary but mean reverting for $0.5\leq d_{i}<1$. As such, the subjects with an estimation of $d_{i}\approx 1$ in at least one condition were removed given that the series is possibly non mean reverting; three subjects were removed, so that a total of 59 subjects was considered for the statistical analysis.

### 3.3. Statistical Analysis

Significant changes in complexity across the pairs of experimental conditions SU${}_{1}$ vs. UP and SU${}_{2}$ vs. MA were assessed via a linear mixed-effects model, that is, a linear regression model that incorporates both fixed and random effects [35]. In our case, the fixed-effects (or factors) were condition and scale, while the random-effect was the subject-dependent intercept that allows for the random variation between subjects. Additionally, the interaction between the factors is also considered. The significance of both the effects (fixed and random) and the interaction between fixed effects was assessed by the significance of the corresponding estimated coefficients at the level of p<0.05. Residuals were checked for whiteness.

To evaluate the changes of interest, estimated marginal means (EMM) [36] were computed for each difference, or contrast, UP-SU${}_{1}$ and MA-SU${}_{2}$, at each frequency of interest (0.4 Hz, 0.15 Hz, 0.1 Hz and 0.04 Hz). A Z-test was applied to check the significance of these differences with a significance level of p<0.05 and Tukey correction for multiple comparisons. The packages lme4 [37] and emmeans [38] of the R software [39] were used to build the model and to compute EMM, respectively.

## 4. Results

This section presents the results of multiscale analysis performed for the multivariate complexity ${\bar{C}}_{X}$ as defined in Equation (Equation 3), as well as for the complexity computed for the scalar process $X_{HP}$ in four different ways: the univariate complexity of HP with respect to its own dynamics, ${\bar{C}}_{X_{HP}|X_{HP}}$ (Equation ([Disp-formula FD4a-entropy-22-00315])), the bivariate complexity of HP with respect to its own dynamics and to the dynamics of SAP, ${\bar{C}}_{X_{HP}|X_{SAP},X_{HP}}$, or RESP, ${\bar{C}}_{X_{HP}|X_{RESP},X_{HP}}$ (Equation (4b)), and the multivariate complexity of HP with respect to the dynamics of the whole trivariate process, ${\bar{C}}_{X_{HP}|X}$ (Equation (4c)).

Figure 1 presents the median and quartiles across subjects of the multivariate complexity ${\bar{C}}_{X}$ computed for eVAR (first row) and eVARFI (second row) as a function of the time scale τ=1,⋯,30, for SU${}_{1}$ vs. UP (left column) and SU${}_{2}$ vs. MA (right column). The measure always tends to its theoretical value $\frac{1}{2}\ln{\left(2\pi e\right)}^{3}$ when computed for eVAR at high values of τ. In fact, the complexity value is in general lower for eVARFI than for eVAR, and presents more variability across subjects. From a visual inspection of the multiscale patterns one can infer that for eVAR there is a decrease in complexity moving from SU${}_{1}$ to UP, evident at short time scales (τ<10), and that the complexity seems not to change while moving from SU${}_{2}$ to MA. For eVARFI, the decrease from SU${}_{1}$ to UP is visible across the whole range of time scales and there seems to be a decrease from SU${}_{2}$ to MA for scales larger than 4.

Figure 2 presents the same as the previous figure but for the univariate (Figure 2a), bivariate (Figure 2b with SAP and Figure 2c with RESP) and multivariate (Figure 2d) complexity measures relevant to the HP time series. Again, eVARFI estimation presents more variability than eVAR, visible across all measures, and does not reach the theoretical value $\frac{1}{2}\ln\left(2\pi e\right)$ at long time scales. Although the four measures are similar with each other, slight changes occur when the complexity of HP is computed accounting for the dynamics of the other time series. When compared to the univariate measure ${\bar{C}}_{X_{HP}|X_{HP}}$, the median complexity value decreases for the bivariate measures at lower time scales, being lower with RESP (${\bar{C}}_{X_{HP}|X_{RESP},X_{HP}}$) than with SAP (${\bar{C}}_{X_{HP}|X_{SAP},X_{HP}}$); the multivariate measure ${\bar{C}}_{X_{HP}|X}$ presents even lower median complexity values for lower time scales. For both eVAR and eVARFI, the complexity of HP decreases at short time scales for all four measures. The differences are more subtle during MA, where only a slight decrease in the median is noticeable at very short time scale for eVARFI.

Figure 3 reports the distribution of the multivariate complexity measure ${\bar{C}}_{X}$ (blue dots) (values and boxplot) computed at the four predetermined cutoff frequencies of the multiscale filter (0.4 Hz, 0.15 Hz, 0.1 Hz and 0.04 Hz) for each experimental condition. Statistically significant changes (p<0.05) in complexity across the pairs SU${}_{1}$ vs UP or SU${}_{2}$ vs MA, as assessed by the estimated marginal means based on the linear mixed-effects model, are marked with *. Comparing SU and UP, the multivariate complexity decreases significantly both for eVAR and eVARFI at 0.4 Hz, while no significant differences are observed at lower cutoff frequencies. Moving from SU${}_{2}$ to MA, the measure increases significantly for eVAR at all frequencies except 0.04 Hz, but decreases for eVARFI at frequencies 0.1 Hz and 0.04 Hz.

Figure 4 presents the same information as the previous figure but considering the HP only as target process and computing the corresponding univariate (Figure 4a), bivariate (Figure 4b with SAP and Figure 4c with RESP) and multivariate (Figure 4d) complexity measures. We find that, moving from SU to UP, the univariate measure ${\bar{C}}_{X_{HP}|X_{HP}}$ evaluated at a time scale corresponding to the cutoff frequency of 0.4 Hz decreases significantly for both eVAR and eVARFI; this decreased complexity of HP in the UP condition is observed identically when the dynamics of SAP, of RESP, and of both SAP and RESP are included in the conditional entropy measure. Moreover, the inclusion of one or both the other time series makes such a decrease statistically significant also with a cutoff of 0.15 Hz when eVAR analysis is performed. Moving from SU${}_{2}$ to MA, we observe that only the eVARFI analysis detects statistically significant differences. Specifically, using eVARFI the univariate complexity of HP decreases during MA at time scales compatible with the cutoff frequency of 0.4 Hz, and also with cutoff 0.15 Hz when SAP dynamics (but not RESP dynamics) are considered. For both protocols and using both estimation methods, the analysis at longer time scales (cutoff 0.1 Hz and 0.04 Hz) does not lead to any significant variation in any of the HP complexity measures.

Figure 5 depicts the distribution across subjects of the differencing parameter $d=[d_{HP},d_{SAP},d_{RESP}]$ computed using the Whittle semiparametric estimator for the three time series and for each condition. The differencing parameter is positive for HP and SAP, while it is lower and centered around zero for RESP. The statistical significance of the variations of this parameter moving from SU${}_{1}$ to UP, or from SU${}_{2}$ to MA, was tested with the same linear mixed-effects model used for the complexity measures. Results indicate that there are statistically significant differences in the values of *d* only for SAP, documenting that conditions of stress evoked by UP or MA determine an increase of the differencing parameter estimated for this time series.

## 5. Discussion

Building on recent developments regarding linear multiscale time series analysis [7,8,19,20], the present study introduces a novel approach to assess the dynamical complexity of multivariate processes accounting for the presence of short term dynamics and long-range correlations. The adopted linear parametric framework retains the advantage of previous formulations [7,19,20] related to the computational reliability of complexity estimated even at coarse time scales and over relatively short time series (few hundred points), and is here integrated with the incorporation of long-range dynamics (fundamental to a proper evaluation of complexity at coarse time scales [8]) and extended for the first time towards a fully multivariate implementation.

The usefulness of the proposed multivariate and multiscale measure of complexity was demonstrated in practice by the analysis of cardiovascular and cardiorespiratory interactions. In particular, the multivariate measure of complexity allowed us to evaluate the overall impact that the physiological mechanisms related to postural and mental stress have on the joint dynamics of HP, SAP and RESP. Simultaneously, the multiscale assessment of complexity led to elicit the contribution of mechanisms operating across multiple time scales (evaluation at fine time scales, with cutoff frequency of 0.4 Hz encompassing VLF, LF and HF oscillations) and that of mechanisms confined to slower oscillations (evaluation at coarser time scales, with cutoff at 0.15 Hz excluding HF oscillations and cutoffs at 0.1 Hz and 0.04 Hz excluding progressively also LF oscillations). In addition the evaluation of the complexity of HP, either considering its own dynamics only or the dynamics of SAP and/or RESP, was exploited to relate the overall complexity trends reflected in the multivariate measure to variations specifically related to physiologically well-known mechanisms of cardiovascular and cardiorespiratory interactions. To aid interpretation of the results and the discussion of the related physiological mechanisms, we report in Table 1 the statistically significant increase or decrease in complexity observed without (eVAR method) or with (eVARFI method) modeling long-range correlations during head-up tilt (comparison SU${}_{1}$ vs. UP) or during mental arithmetics (comparison SU${}_{2}$ vs. MA) at the time scales encompassing the whole spectral content (0.4 Hz), filtering the HF oscillations (0.15 Hz), and filtering in part (0.1 Hz) or completely (0.04 Hz) the LF oscillations. Complexity is linearly dependent on tilt inclination: a steeper inclination of tilt table produces higher degrees of sympathetic activation, with corresponding lower values of entropy-based complexity measures [40]. Studying the trends of the multivariate complexity measure, its decrease with tilt (Figure 3) documents a simplification of the overall dynamics of HP, SAP and RESP. This result can be mainly driven by the less complex dynamics of heart rate variability in the upright position, that we document calculating the complexity of HP (Figure 4a). Indeed it is well known that the dynamics of HP tend to become less complex as a consequence of the rise of LF oscillations and the weakening of HF oscillations related to the shift of the sympathovagal balance towards sympathetic activation and vagal deactivation during head-up tilt [15,40]. The fact that the decrease of multivariate complexity is not statistically significant for cutoff 0.15 Hz and lower suggests that it is related mainly to the weakening of HF oscillations; indeed, HF oscillations of both HP and SAP are known to be blunted with tilt [15,41,42]. In addition, the finding that the decrease in complexity is observed in the same way for VAR and VARFI analysis suggests that the impact of long-range correlations does not change substantially from rest to tilt. Overall, these results suggest that cardiovascular and respiratory oscillations in the LF and VLF band considered together do not alter significantly their complexity in the transition from the supine to the upright body position.

On the other hand, the increase of the global complexity with MA observed for eVAR modeling across multiple time scales (cutoffs 0.4, 0.15, 0.1) is in agreement with previous findings relevant to the HP time series [7]. The fact that such increase is not observed for eVARFI indicates that long range correlations have a different impact on the cardiovascular and respiratory dynamics during MA compared with the relaxed condition SU${}_{2}$. In particular, eVARFI estimation reports values of the multivariate complexity which are unchanged at low time scales (cutoff 0.4 Hz), tend to decrease at cutoff 0.15 Hz, and decrease significantly at cutoffs 0.1 Hz and 0.04 Hz. We ascribe this lower complexity to a larger contribution of long-range correlations acting in LF and especially VLF bands, which is supported by the fact that the differencing parameter increases significantly, for HP and especially for SAP, during MA. This confirms the regularizing role of long range correlations on physiological dynamics [8,10].

The complexity of HP assessed at lower time scales (cutoff 0.4 Hz) always decreases in the UP position, for all measures (univariate, both bivariate, and multivariate), as shown in Figure 4. This documents the well-known simplification of heart rate variability induced by head-up tilt, which is known to evoke sympathetic activation and vagal withdrawal making the cardiac dynamics more regular [31,40,43,44]. The fact that this result is observed identically for the eVAR and eVARFI approaches indicates that long-range correlations do not impact significantly the evaluation of complexity performed at short time scales. On the other hand, focusing on intermediate time scales for which HF oscillations are cut off ($f_{\tau }$ = 0.15 Hz), we find that the decrease in complexity is lost when considering the HP dynamics individually, but is maintained when SAP and RESP, taken individually or together, are used to reduce the complexity of HP; moreover, this holds for eVAR but not for eVARFI. Taken together, these results suggest that slow oscillations of SAP and/or RESP reduce the complexity of HP in the transition from rest to tilt, and such regularizing action is related to long-range correlations. This finding is compatible with the known increase of cardiovascular interactions during tilt, documented in previous studies using a number of causality measures including information-theoretic ones [13,24,26,27], and also with the existence of slowly varying respiratory patterns that enhance their effect on the variability of HP during postural stress [24]. Here, the presence of cardiorespiratory and cardiovascular interactions is documented indirectly observing the lower values during tilt of the bivariate and multivariate complexity measures compared with the univariate one, through an approach similar to that proposed in Ref. [14].

The complexity of HP tends to decrease also with the mental stress induced by MA. Contrary to the postural stress induced by HUT, the decrease is observed only using the VARFI approach, and not using the VAR. The absence of significant changes in complexity using methods that do not model long-range correlations is in agreement with previous findings [7,14]. Our results document that the decrease in complexity is due to the different impact of long-range correlations, detected only modeling them through the VARFI approach, and again are confirmed by the higher values of the differencing parameter estimated for HP and for SAP (but indeed not for RESP) during MA compared with the relaxed condition (Figure 5). A differencing parameter *d* = 0 indicates that there are no long-range correlations. The rather small values of *d* observed for the RESP time series likely reflect the fact that, even if not controlled, the respiratory activity is confined in a narrow band of the frequency spectrum and thus it does not exhibit the slow trends typical of long-range correlated dynamics. Moreover, the changes of the differencing parameter *d* observed in the upright position for the systolic pressure (and only to a smaller extend for the heart period) document that the sympathetic activation produced by the tilt table inclination may modulate the impact that long range correlations have on the cardiovascular variability. We note also that, when SAP is considered, the significant changes extend to the second cutoff frequency (0.15 Hz), indicating the increasing contribution of SAP long-memory dynamics in reducing the complexity of HP. Therefore, we state the importance of accounting for long-range correlations in the assessment of the changes in the complexity of heart rate variability induced by mental stress. This may have relevance for the practical applications focused on the detection of mental workload or stress [45,46,47,48]).

Future extensions of the present work can be targeted first at investigating methodological aspects, such as the possibility of describing interactions between processes at the level of long-range correlations (possibly with the modeling of cointegration [49]) or the extension of the framework to the direct evaluation of Granger-causal interactions [20] in a multivariate context where long-range correlations are modeled [8]. Regarding applicative contexts, the analysis of multivariate multiscale dynamics is of particular interest in econometrics [50] and in the neurosciences [51,52], where dynamics spanning several temporal scales are commonly observed and multichannel data acquisition is ubiquitous.

## Figures and Tables

**Figure 1 entropy-22-00315-f001:**
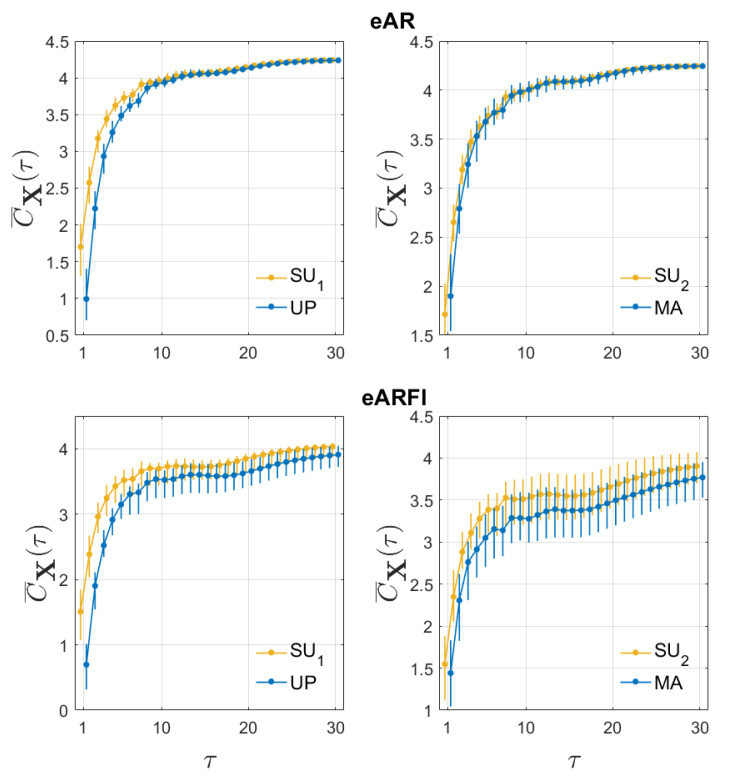
Distribution across subjects of the multivariate complexity measure ${\bar{C}}_{X}$ as a function of the time scale τ for eVAR (first row) and eVARFI (second row), for SU${}_{1}$ vs. UP (left column) and SU${}_{2}$ vs. MA (right column).

**Figure 2 entropy-22-00315-f002:**
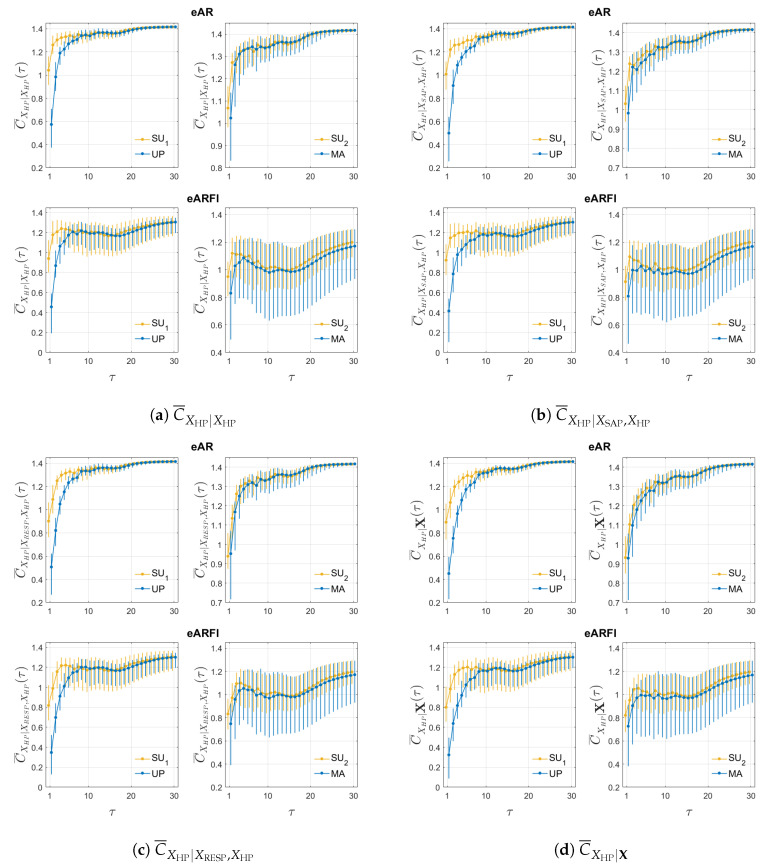
Distribution across subjects of the univariate (**a**), bivariate (**b**) with SAP and (**c**) with RESP, and multivariate (**d**) complexity measures as a function of the time scale τ for eVAR (first row) and eVARFI (second row), for SU1 vs. UP (left column) and SU2 vs. MA (right column).

**Figure 3 entropy-22-00315-f003:**
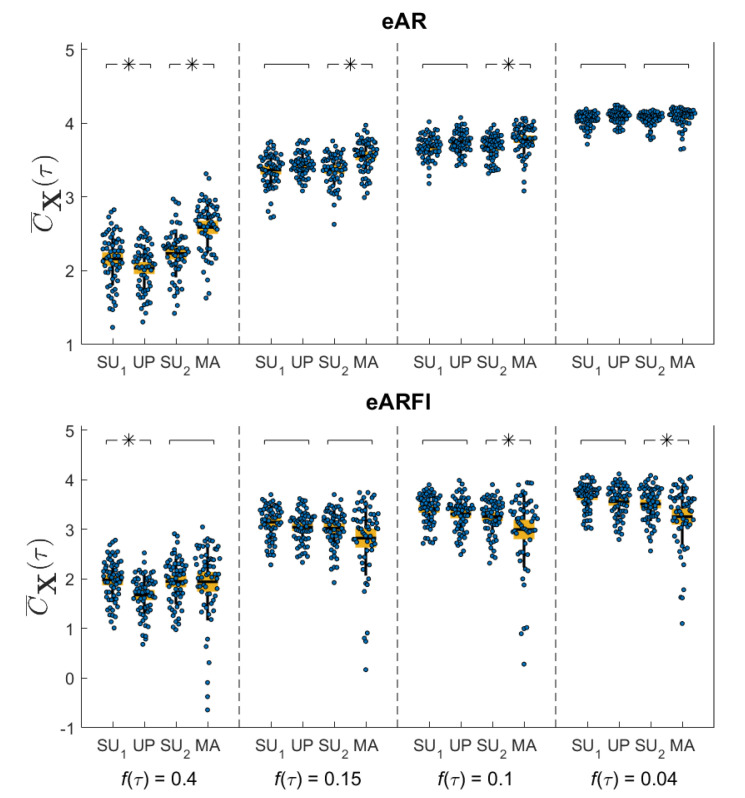
Distribution of the multivariate complexity measure ${\bar{C}}_{X}$, depicted as boxplot (mean and confidence intervals, yellow filled box; standard deviation, black vertical line) and original values (dots) for selected frequencies ($f_{\tau_{Hz}}=$ 0.4 Hz; 0.15 Hz; 0.1 Hz; 0.04 Hz), computed for the four experimental conditions using eVAR (first row) and eVARFI (second row) identification methods. Statistically significant differences between pairs of conditions are marked with an asterisk.

**Figure 4 entropy-22-00315-f004:**
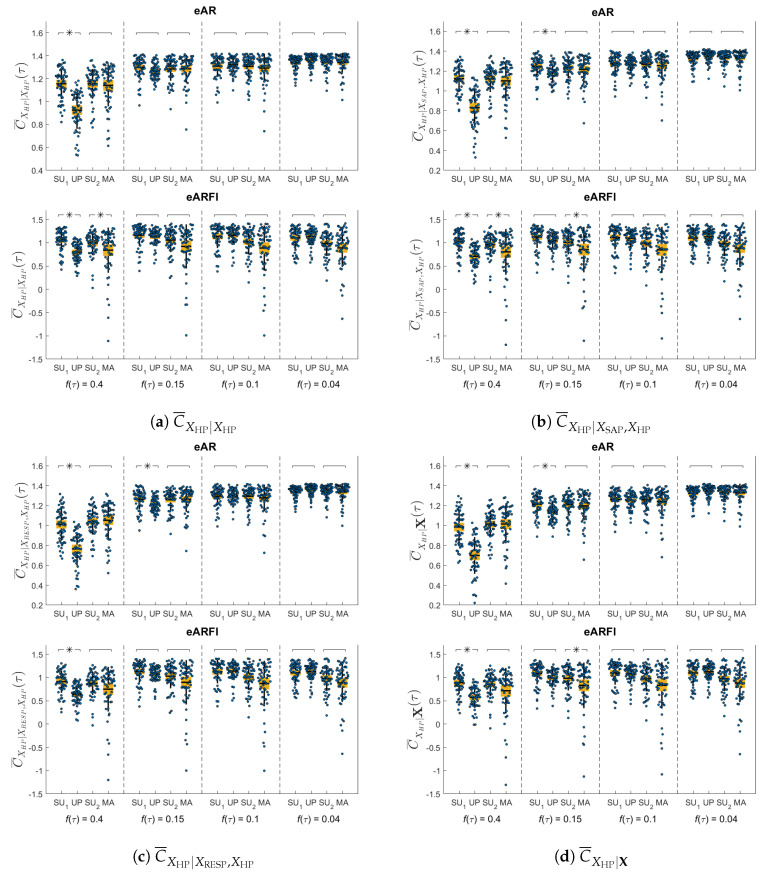
Distribution of the univariate (**a**), bivariate (**b**) with systolic arterial pressure (SAP) and (**c**) with respiration (RESP), and multivariate (**d**) complexity measures depicted as boxplot (mean and confidence intervals, yellow filled box; standard deviation, black vertical line) and original values (dots) for selected frequencies (ftHz = 0.4 Hz; 0.15 Hz; 0.1 Hz; 0.04 Hz), computed for the four experimental conditions using eVAR (first row) and eVARFI (second row) identification methods. Statistically significant differences between pairs of conditions are marked with an asterisk.

**Figure 5 entropy-22-00315-f005:**
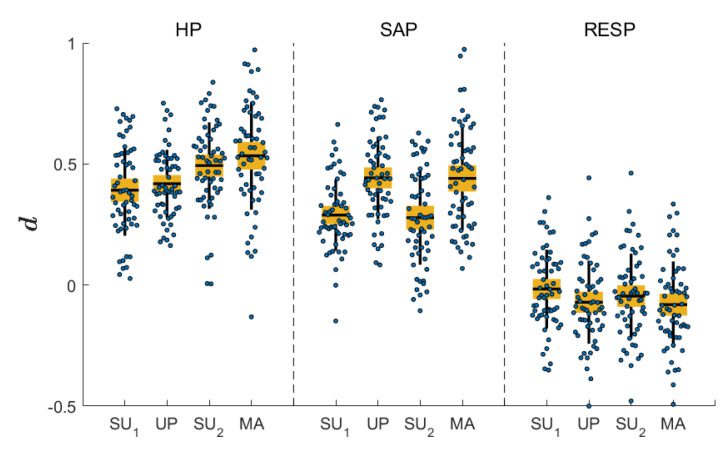
Distribution of the long-rang parameter $d_{i}$ for each of the time series considered (heart period (HP), SAP and RESP) and for the four conditions.

**Table 1 entropy-22-00315-t001:** Significant differences (*p*-value < 0.05) between pair of conditions for each measure and frequency. The arrows indicate if the measure increases or decreases from rest to stress.

Measure	Approach	SU1 → UP	SU2 → MA
0.4	0.15	0.1	0.04	0.4	0.15	0.1	0.04
${\bar{C}}_{X}$	eVAR	↘				↗	↗	↗	
eVARFI	↘						↘	↘
${\bar{C}}_{X_{HP}|X_{HP}}$	eVAR	↘							
eVARFI	↘				↘			
${\bar{C}}_{X_{HP}|X_{SAP},X_{HP}}$	eVAR	↘	↘						
eVARFI	↘				↘	↘		
${\bar{C}}_{X_{HP}|X_{RESP},X_{HP}}$	eVAR	↘	↘						
eVARFI	↘							
${\bar{C}}_{X_{HP}|X}$	eVAR	↘	↘						
eVARFI	↘					↘

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
