# Peer review of "Multivariate and Multiscale Complexity of Long-Range Correlated Cardiovascular and Respiratory Variability Series"

_entropy, 2020, doi:10.3390/e22030315_

Round 1

Reviewer 1 Report

The article “Multivariate and Multiscale Complexity of Long-Range Correlated Cardiovascular and Respiratory Variability Series” proposes a method to compute the multiscale complexity of a multivariate time series where there is a coexistence of short-term dynamics and long-range correlation. The method computes the multiscale complexity of the complete multivariate process or just for one of the scalar process. The former is achieved in terms of the variance of the innovations and downsampled multivariate process based on a vector autoregressive fractionally integrated (VARFI) model and, in term of a state-space theory, the changes of the VARFI model parameters with the scale of observation. The paper is well-written and organized. It provides all the steps and assumptions that are necessary, besides the appendices are good complement of the paper. The method is applied to data from a head-up tilt test and mental arithmetic stress that only involves healthy subjects. Just a few questions regarding the application:

  • In general, there are some parameters that do not have an associated explanation about their value. For example, the authors indicate that “q” in equation 7a was fixed to 50. Is it possible to add some discussion about this value? How it affects the multiscale complexity results?
  • Regarding the order of the FIR filter to produce the different time scales, is it not to low?
  • The complexity curves in figure 2 provided by the eAR and eARFI are pretty close to each other for the head-up tilt test. Could this be related to the fact that the tilt was only 45 degrees? Also, in figure 5 the “d” value did not change for HP from SU1 to UP and just a little for SAP.
  • How to understand that for RESP the mean value of “d” in figure 5 is zero? Were the subjects breathing at certain breathing frequency?

Reviewer 2 Report

I have read the paper by Martins et. al. with great pleasure. This is a very well written and laid out paper, with sound methods and results. The authors set out to solve the notorious problem of MSE, which is its unsuitability in the presence of short multivariate time series to be analyzed at long time scales. They introduce new methods to address this problem. 

The paper is a bit mathematically dense and requires some mathematical sophistication, so the authors might consider more explanations for the notation used. Also, the paper is not a purely theoretical study, so the authors should mention that they actually analyze data in the Abstract. 

Other than that I think that the paper is great and should be published in Entropy. I recommend it for publication without any doubts.

Reviewer 3 Report

This work presents a new method for the assessment of the multiscale complexity of multivariate time series, which is functionally depicted analyzing heart period, the systolic arterial pressure, and the respiration—demonstrating the advantages of this method respect to multiscale entropy. The new method is based on vector autoregressive fractionally integrated (VARFI) models to yield a linear parametric, short- and long-range correlations and state-space models.

The experiments were well conducted, and some minor issues should be attended:

  1. line 265 correct word "ohysiological".
  2. I suggest depicting the new method exampling its application using a basic series, and present the method using algorithm form.
